# FINE-GRAINED CONTRASTIVE LEARNING FOR ECG-REPORT ALIGNMENT WITH WAVEFORM ENHANCEMENT

## ABSTRACT

Electrocardiograms (ECGs) are essential for diagnosing cardiovascular diseases. However, existing ECG-Report contrastive learning methods focus on whole-ECG and report alignment, missing the link between local ECG features and individual report tags. In this paper, we propose FG-CLEP (Fine-Grained Contrastive Language ECG Pre-training), which achieves fine-grained alignment between specific ECG segments and each tag in the report via tag-specific ECG representations. Furthermore, we found that nearly 55% of ECG reports in the MIMIC-ECG training dataset lack detailed waveform features, which hinders fine-grained alignment. To address this, we introduce a coarse-to-fine training process that leverages large language models (LLMs) to recover these missing waveform features and validate the LLM outputs using a coarse model. Additionally, fine-grained alignment at the tag level, rather than at the report level, exacerbates the false negative problem, as different reports may share common tags. To mitigate this, we introduce a semantic similarity matrix to guide the model in identifying and correcting false negatives. Experiments on six datasets demonstrate that FG-CLEP significantly improves fine-grained alignment, outperforming state-of-the-art methods in both zero-shot prediction and linear probing. Meanwhile, the fine-grained reports we generate also enhance the performance of other methods. Our code and data are available at: https://anonymous.4open.science/r/FG-CLEP-3454.

## 1 INTRODUCTION

Electrocardiograms (ECGs) are essential non-invasive tools for detecting cardiac rhythm disorders in clinical practice (Sahoo et al., 2020; Rath et al., 2021; Ayano et al., 2022). Recently, self-supervised learning (SSL) has emerged as a promising paradigm for ECG representation learning, alleviating the reliance on large-scale annotated data and expert knowledge. Existing ECG SSL approaches can be broadly categorized into comparative methods (Chen et al., 2020; 2021; Wang et al., 2023; Eldele et al., 2021) and generative methods (Zhang et al., 2022a; Hu et al., 2023; Na et al., 2024; Zhang et al., 2022b). However, most of these methods still struggle with unseen classes in zero-shot scenarios.

Inspired by the success of multimodal contrastive learning such as CLIP (Radford et al., 2021), recent studies have explored ECG–report alignment to enable zero-shot prediction. For instance, Li et al. (2024); Liu et al. (2024a); Lalam et al. (2023) align ECG signals with paired textual reports, while MERL (Liu et al., 2024b) enhances such representations with uni-modal alignment and descriptive prompts from LLMs. ESI (Yu et al., 2024) further integrates retrieval-augmented generation (RAG) (Ni et al., 2025; Gao et al., 2023) pipelines to enrich ECG reports with external medical knowledge. These efforts demonstrate the potential of ECG–text alignment for zero-shot learning.

Despite recent advances, current methods predominantly focus on aligning entire ECGs with their corresponding reports, neglecting the fine-grained relationship between local ECG features and individual report tags. To achieve fine-grained alignment, we identify and address three key challenges.

**1.Fine-grained Alignment Architecture**: Most existing approaches align entire ECGs with whole reports, but they fail to capture patch-level ECG embeddings and tag-specific report embeddings, both of which are critical for fine-grained alignment.

**2.Missing Waveform Features in Reports**: We observe that nearly 55% of ECG reports in the MIMIC-ECG dataset—one of the largest ECG-Report datasets—lack detailed waveform features, which hampers fine-grained alignment. In clinical practice, physicians often begin by identifying key waveform patterns in an ECG before formulating a diagnosis. However, many physicians do not explicitly record these features in the reports, resulting in a significant portion of reports lacking important waveform information. Recovering these missing features using large language models (LLMs) is challenging for two main reasons: (1) the hallucination problem inherent in LLMs (Huang et al., 2023; Günay et al., 2024; Zhang et al., 2025), and (2) the non-bijective relationship between waveform features and diagnostic outcomes, where the same disease may manifest with different waveform patterns (Jin, 2018). As a result, relying solely on LLMs to augment reports, as attempted by Yu et al. (2024), proves to be unreliable.

**3.False Negative Challenge**: Fine-grained alignment at the tag level, rather than at the report level, exacerbates the false negative problem. This is because different reports may share common tags, leading to potential misalignments.

In this study, we propose FG-CLEP to address the aforementioned challenges. Rather than aligning the entire ECG embedding with the report embedding, we perform alignment at the ECG patch level and the individual tag level in the report. Specifically, we obtained tag-specific ECG representations, where different tags in the report serve as queries to attend to ECG patches (treated as keys and values) through cross-attention. To address missing waveform features in reports and generate fine-grained reports, we propose a coarse-to-fine training process. First, we train a coarse CLEP model using contrastive learning on the original ECG-report pairs. Then, we use LLMs to generate potential waveform features from the reports, which are validated using CLEP. Finally, we integrate these validated features into the reports and continue training the CLEP model to obtain the final FG-CLEP model. This approach resolves the non-bijective relationship between waveform features and diagnoses and corrects errors from LLM hallucinations. Lastly, to address the false negative problem exacerbated by fine-grained tag-level alignment, we introduce a semantic similarity matrix. This matrix computes the similarity between tags and is used during contrastive learning to guide the model in identifying and correcting false negatives.

We validate our proposed FG-CLEP on six ECG multi-label classification datasets in both zero-shot and linear probing, the results demonstrate that FG-CLEP significantly improves fine-grained alignment, outperforming state-of-the-art methods in both zero-shot prediction and linear probing. Meanwhile, the fine-grained reports we generate also enhance the performance of other methods. Overall, our contributions are threefold:

- We propose a new alignment architecture, enabling fine-grained alignment between ECG segments and report tags through tag-specific ECG representations, capturing detailed ECG-report relationships.

- We introduce a coarse-to-fine training process using LLMs to recover missing waveform features and validate them with a coarse CLEP model, addressing non-bijective relationships and LLM hallucinations.

- We present a semantic similarity matrix to mitigate false negatives in ECG-tag pairs, guiding contrastive learning to correct misalignments.

- Experimental results show that FG-CLEP, pre-trained on MIMIC-ECG, outperforms state-of-the-art methods in zero-shot prediction and linear probing across six datasets, including PTB-XL, CPSC2018, and CSN.

## 2 RELATED WORK

**ECG-Report Contrastive Learning** Recently, inspired by the strong zero-shot ability of image-caption multimodal contrastive learning methods like CLIP (Radford et al., 2021), significant efforts have been made in ECG-Report contrastive learning (Li et al., 2024; Liu et al., 2024b;a; Yu et al., 2024; Lalam et al., 2023). Similar to CLIP (Radford et al., 2021), Li et al. (2024); Liu et al. (2024a); Lalam et al. (2023); Wang et al. (2025); Pham et al. (2024) learns ECG representations by pulling ECGs with their paired reports while pushing them from unpaired reports. MERL (Liu et al., 2024b) further introduces uni-modal alignment and employs the CKEPE pipeline at inference to generate more descriptive prompts via LLMs. However, enhancing textual prompts only during

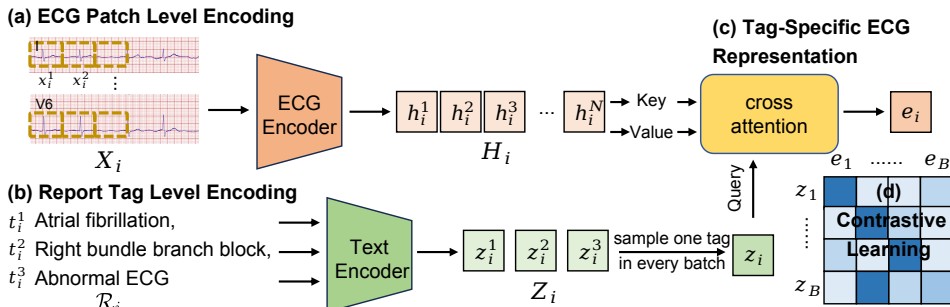

Figure 1: Fine-grained Alignment Architecture with Tag-Specific ECG Representation.

inference creates a distribution mismatch between training and testing text. In contrast, ESI (Yu et al., 2024) enhances ECG reports during training using a retrieval-augmented generation (RAG) pipeline, integrating LLMs and external medical knowledge for more detailed descriptions.

Despite these advances, existing methods all perform contrastive learning at the whole-ECG and report level, and overlook the absence of fine-grained waveform features in ECG reports. To address these challenges, we propose the FG-CLEP.

**Fine-Grained Contrastive Learning** Recent studies in medical imaging have demonstrated the benefits of fine-grained alignment. Methods such as MedFILIP (Liang et al., 2025), fVLM (Shui et al., 2025) incorporate entity-level or region-level supervision by leveraging structured information extracted from reports or by segmenting images into anatomical regions. These approaches enable more precise matching between local visual features and textual descriptions, resulting in improved downstream performance, especially in zero- and few-shot settings.

However, the application of fine-grained contrastive learning in the ECG domain remains largely unexplored. To bridge this gap, we propose to model fine-grained correspondences between characteristic ECG waveform segments and specific textual patterns in the reports, aiming to capture subtle diagnostic cues that may be overlooked in global-level alignment.

**False Negatives in Contrastive Learning** Traditional multi-modal contrastive learning (Radford et al., 2021) assumes that only images and captions from the same record are positive pairs. However, this assumption often fails in the ECG domain, where most ECGs are normal, and abnormalities typically involve common diseases, leading to frequent false negatives. Furthermore, fine-grained alignment at the tag level exacerbates this issue, as different reports may share common tags. There have been several attempts to address this issue (Lavoie et al., 2024; Jiang et al., 2023b; Sun et al., 2023; Li et al., 2023; Kim et al., 2025). Some approaches (Jiang et al., 2023b; Li et al., 2023) attempt to add a regularization term to mitigate false negatives. Others (Sun et al., 2023; Wang et al., 2022) introduce a matrix to measure the similarity between different reports, guiding contrastive learning to identify and address false negatives. In this paper, fine-grained alignment introduces a more pronounced false negative problem, as different reports may share common tags.

## 3 METHOD

### 3.1 FINE-GRAINED ALIGNMENT

In this section, we describe how we achieve fine-grained alignment between specific ECG segments and tags in clinical reports. Given an ECG-report dataset $\mathcal{D} = \{(\mathbf{X}_i, \mathcal{R}_i)\}_{i=1}^{M}$, each report $\mathcal{R}_i$ consists of multiple clinically meaningful tags, such as *arrhythmia*, *myocardial infarction*, *atrial fibrillation*, etc. For each tag, our objective is to identify and align the corresponding ECG segment(s) that reflect this clinical finding. To achieve this, we first perform **ECG patch-level encoding** and **report tag-level encoding** to obtain fine-grained representations of both the ECG signals and the report tags.

**ECG Patch-Level Encoding.** To obtain fine-grained ECG representations, we employ ViT to encode patch-level features from the ECG signal. Given a 12-lead ECG $\mathbf{X}_i \in \mathbb{R}^{L \times T}$, where $L = 12$ is the number of leads and $T$ is the signal length, we independently divide each lead into $N_{\text{lead}}$ non-

overlapping segments (patches) along the temporal axis, with each patch of length $\Delta T = T/N_{\text{lead}}$. This yields a total of $N = L \times N_{\text{lead}}$ patches:

$$\mathbf{X}_i \rightarrow \{\mathbf{x}_i^p\}_{p=1}^N, \quad \mathbf{x}_i^p \in \mathbb{R}^{1 \times \Delta T}. \tag{1}$$

All $N$ patches are passed through the ECG encoder resulting in a sequence of patch-level embeddings:

$$\mathbf{H}_i = \left[\mathbf{h}_i^1, \ldots, \mathbf{h}_i^N\right] = \text{ECGEncoder}\left(\left[\mathbf{x}_i^1, \ldots, \mathbf{x}_i^N\right]\right) \in \mathbb{R}^{N \times d}. \tag{2}$$

**Report Tag-Level Encoding.** Since ECG reports are highly structured, we can simply use commas to split the tags. We denote the set of tags in the $i$-th report as $\mathcal{T}_i = \{t_i^j\}_{j=1}^{m_i}$, where $t_i^j$ represents the $j$-th tag in report $i$, and $m_i$ is the number of tags in $\mathcal{R}_i$. Each tag $t_i^j$ is then independently encoded via a text encoder to obtain its embedding representation. This yields the tag-level representation matrix for report $i$:

$$\mathbf{Z}_i = [\mathbf{z}_i^1, \ldots, \mathbf{z}_i^{m_i}] \in \mathbb{R}^{m_i \times d}, \tag{3}$$

$$\mathbf{z}_i^j = \text{TextEncoder}(t_i^j), \quad j = 1, \ldots, m_i. \tag{4}$$

Notably, both the ECG patch embeddings and tag embeddings are projected into the same latent space with dimension $d$. For clarity and conciseness, we omit explicit projection formulas in the equations above.

**Tag-Specific ECG Representation.** With the above fine-grained ECG patch embeddings and report tag embeddings, we can now align them at a fine-grained level. Ideally, if explicit annotations mapping each tag to its corresponding ECG patches were available, direct supervised alignment could be applied. However, such fine-grained annotations are typically unavailable in practice. To address this, we propose an automatic alignment mechanism based on cross-attention.

Specifically, for each tag, we use its embedding as the query and the ECG patch embeddings as keys and values. Through the cross-attention mechanism, the model adaptively computes a tag-specific ECG representation by attending to ECG patches according to their relevance to the tag. This enables each tag to aggregate information from the most relevant ECG segments, capturing the fine-grained relationship without requiring extra supervision.

Formally, given a tag embedding $\mathbf{z}_i^j$ and ECG patch embeddings $\mathbf{H}_i$, we compute the tag-specific ECG representation as:

$$\mathbf{e}_i^j = \text{CrossAttn}(\mathbf{z}_i^j, \mathbf{H_i}) = \sum_{p=1}^N \alpha_{j,p} \cdot \mathbf{h}_i^p, \tag{5}$$

$$\alpha_{j,p} = \frac{\exp(\langle \mathbf{z}_i^j, \mathbf{h}_i^p \rangle)}{\sum_{p'=1}^N \exp(\langle \mathbf{z}_i^j, \mathbf{h}_i^{p'} \rangle)}. \tag{6}$$

**Fine-Grained Contrastive Learning Objective.** For each random batch with size $B$, we sample one tag $z_i$ from each report's tag set $\mathbf{Z}_i$, and compute its corresponding tag-specific ECG representation $\mathbf{e}_i$. The learning objective is to maximize the similarity between each tag embedding and its corresponding ECG representation, while minimizing the similarity with unpaired ones, similar to SigLIP (Zhai et al., 2023) for its efficiency compared to the original CLIP.

$$L_{\text{con}} = -\frac{1}{B} \sum_{i=1}^B \sum_{j=1}^B \log\left(\frac{1}{1 + \exp\left(-y_{ij} \cdot t \cdot \text{sim}(\mathbf{e}_i, \mathbf{z}_j)\right)}\right). \tag{7}$$

where $y_{ij}$ denotes the match between a given ECG and report input(1 if i==j, otherwise -1), $\text{sim}(\cdot, \cdot)$ denotes the cosine similarity, and $t$ is a temperature hyperparameter.

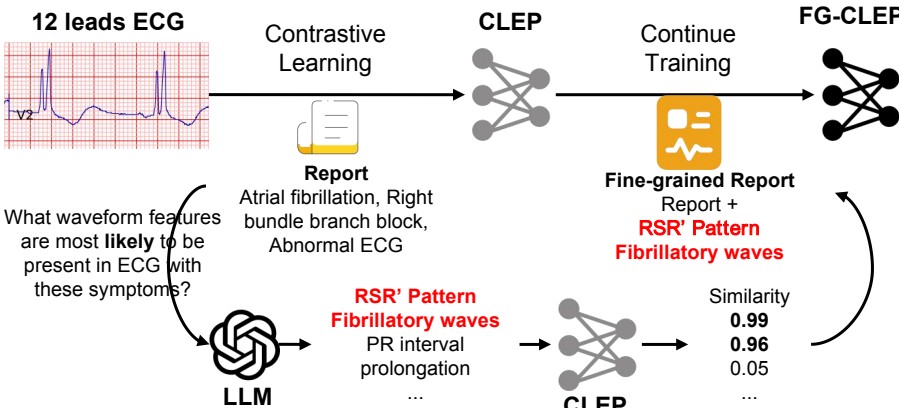

Figure 2: **Coarse-to-Fine Training Process of FG-CLEP.** The reports with recovered waveform features are referred to as fine-grained reports. Models trained with fine-grained reports are prefixed with FG-.

## 3.2 COARSE-TO-FINE TRAINING PROCESS

Fine-grained alignment relies on fine-grained reports that include detailed waveform features. Although we achieve fine-grained alignment between tags and specific ECG segments, as discussed above, we observe that due to clinical habits, nearly half of the reports do not record waveform features as intermediate observations. Therefore, we propose a novel training process illustrated in Figure 2 consisting of three steps: (1) training the CLEP model using contrastive learning on original ECG-report pairs, (2) generating potential waveform features based on the original report using LLMs and validating them with CLEP, and (3) continuing to train the CLEP model with this augmented report containing waveform features to obtain the final FG-CLEP model.

The key to our training process is to recover the missing waveform features in the report. Given a report, we query LLMs with the question, *'What waveform features are most likely to be present in electrocardiograms with these symptoms?'* to identify potential overlooked waveform features. To format the results, we further instruct, *'Organize these waveform features into a Python list, with each item representing a distinct waveform feature.'* Using this explicit chain-of-thought instruction (Wei et al., 2022), we obtain a list of potential waveform features. However, LLM outputs are unreliable for two reasons: First, ECG waveform features and diagnoses are not in a one-to-one correspondence (Jin, 2018)—a single disease may present different waveform characteristics across individuals. Doctors can infer a diagnosis from waveform features, but not the other way around. Second, even if the non-bijective relationship is excluded, the LLM's output is inherently unstable due to hallucination issues (Huang et al., 2023; Günay et al., 2024).

Thus, we validate LLM-generated waveform features by computing their similarity to the ECG signal using coarse CLEP, selecting only high-confidence waveform features for augmentation. These validated features are then incorporated into the original report for continued training of the Fine-Grained CLEP model.

## 3.3 FALSE NEGATIVE MITIGATION

To address the false negative problem exacerbated by tag-level fine-grained alignment, we introduce a semantic similarity matrix that computes the similarity between different tags. This matrix is incorporated into contrastive learning to guide the model in identifying and correcting false negatives.

Formally, the semantic similarity matrix $\mathbf{S} \in \mathbb{R}^{B \times B}$ is defined as follows.

$$\mathbf{S}_{ij} = \text{sim}(\mathbf{e}_i, \mathbf{e}_j) \in [0, 1]. \tag{8}$$

We integrate the semantic similarity matrix into the loss function to guide contrastive learning, following the approach in (Sun et al., 2023). This matrix captures the semantic similarity between tags from different reports, enabling the model to identify and correct false negative samples by ensuring that similar tags are aligned more effectively, even if they belong to different reports.

The loss term $L_{\text{fnm}}$ is defined as:

$$L_{\text{fnm}} = \frac{1}{B} \sum_{i=1}^{B} \sum_{j=1}^{B} |\text{sim}(\mathbf{e}_i, \mathbf{z}_j) - \mathbf{S}_{ij}|. \tag{9}$$

where $B$ is the batch size and $|\cdot|$ denotes the L1 distance.

The final loss function for FG-CLEP is the combination of the contrastive loss and the false negative mitigation loss:

$$L = L_{\text{con}} + \lambda L_{\text{fnm}}. \tag{10}$$

## 4 EXPERIMENTS

### 4.1 DATASETS

We pre-train the FG-CLEP framework using the MIMIC-ECG (Gow et al.) dataset and test it on the PTB-XL (Wagner et al., 2020), CPSC2018 (Liu et al., 2018), and CSN (Zheng et al., 2022) datasets, following the benchmark proposed by (Liu et al., 2024b). All the ECGs in the datasets are 12-lead recordings. The MIMIC-ECG dataset contains nearly 800,000 ECG-report pairs. To improve data quality, we excluded samples with an empty report or reports containing fewer than three words, removed reports without useful information, and discarded ECGs with unexpected situations. Details regarding the train:validation:test split and other dataset-specific information are provided in the Appendix.

### 4.2 IMPLEMENTATION DETAILS

**Pre-training Implementation**: In the pre-training stage, we utilize a randomly initialized ViT model (Dosovitskiy et al., 2020) as the ECG encoder and BioClinicalBERT (Alsentzer et al., 2019) for text encoding. The whole ECG is divided into 60 non-overlapping patches. The AdamW optimizer is selected with a learning rate of $2 \times 10^{-5}$ and a weight decay of $1 \times 10^{-4}$. CLEP is pre-trained for 10 epochs with original reports and FG-CLEP is trained for another 3 epochs with fine-grained reports, using a cosine annealing scheduler for learning rate adjustments and a warmup phase for the first 10% of training steps. A batch size of 100 is maintained. The temperature parameters $t$ are initialized to $\log 10$. The default hyperparameter $\lambda$ is set to 0.5 and the default threshold for selecting high-confidence waveform features is set to 0.95. We use LLaMA3-8B (AI@Meta, 2024) as our LLM to query potential waveform features and use vLLM (Kwon et al., 2023) to speed up inference. All experiments used two NVIDIA A800 80GB GPUs, except LLaMA3-70B ablation, which used four.

**Downstream Task Implementation**: We evaluated the downstream tasks using both zero-shot and linear probe settings. For the zero-shot setting, we froze the entire model and used the text of each category as the prompt. We computed the similarity between the ECG embedding and the category text embedding as the classification probability. Additionally, we employed an ensemble method to enhance zero-shot performance. Specifically, in addition to using the category as text, we also added 'category in lead x' (x represents any of the 12 leads) as text to compute the probability and used the highest probability as the final probability for that category. For linear probing, we kept the ECG encoder frozen and updated only the parameters of a newly initialized linear classifier. We conducted linear probing for each task using 1%, 10%, and 100% of the training data. For all downstream tasks, we used macro AUC as the metric.

### 4.3 ZERO-SHOT ABILITY

The zero-shot results are illustrated in Table 1. Both CLEP and FG-CLEP performed well. A detailed examination of the data reveals that FG-CLEP significantly outperforms CLEP on PTBXL-Form, PTBXL-Rhythm demonstrating that continue training using fine-grained reports substantially enhanced the model's ability to capture local ECG waveform features. This improvement is particularly evident when using the ensemble method, which extends the label text to 12 leads ('label in lead

Table 1: Results of zero-shot classification. ENS: ensemble inference. FG-: trained with fine-grained reports.

| macro AUC | PTB-XL-Super | PTBXL-Sub | PTBXL-Form | PTBXL-Rhythm | CPSC2018 | CSN |
|---|---|---|---|---|---|---|
| METS (Li et al., 2024) | 76.31 | 80.12 | 65.95 | 86.29 | 82.49 | 77.20 |
| FG-METS | 78.12$_{\uparrow 1.81}$ | 82.01$_{\uparrow 1.89}$ | 66.33$_{\uparrow 0.38}$ | 90.12$_{\uparrow 3.83}$ | 86.92$_{\uparrow 4.43}$ | 81.20$_{\uparrow 4.00}$ |
| MERL (Liu et al., 2024b) | 74.20 | 75.70 | 65.90 | 78.50 | 82.80 | 74.40 |
| FG-MERL | 76.70$_{\uparrow 2.50}$ | 78.20$_{\uparrow 2.50}$ | 66.80$_{\uparrow 0.90}$ | 81.00$_{\uparrow 2.50}$ | 85.30$_{\uparrow 2.50}$ | 76.90$_{\uparrow 2.50}$ |
| MELP | 76.20 | 81.20 | 69.10 | 85.40 | 84.20 | 77.60 |
| D-BETA | 76.20 | 75.90 | 66.10 | 88.60 | 80.10 | 76.30 |
| CLEP | 78.01 | 82.41 | 67.96 | 89.48 | 85.94 | 80.88 |
| FG-CLEP | **80.13**$_{\uparrow 2.12}$ | **84.46**$_{\uparrow 2.05}$ | **68.46**$_{\uparrow 0.50}$ | **93.02**$_{\uparrow 3.54}$ | **88.90**$_{\uparrow 2.96}$ | **83.23**$_{\uparrow 2.35}$ |
| CLEP$_{ENS}$ | 76.26$_{\downarrow 1.75}$ | 83.23$_{\uparrow 0.82}$ | 65.71$_{\downarrow 2.25}$ | 89.18$_{\downarrow 0.30}$ | 84.72$_{\downarrow 1.22}$ | 81.93$_{\uparrow 1.05}$ |
| FG-CLEP$_{ENS}$ | **80.55**$_{\uparrow 0.42}$ | 84.42$_{\downarrow 0.04}$ | **71.85**$_{\uparrow 3.39}$ | **93.52**$_{\uparrow 0.50}$ | 87.93$_{\downarrow 0.97}$ | **85.58**$_{\uparrow 2.35}$ |

Table 2: Results of Linear Evaluation.

| Method | PTB-XL-Super | | | PTBXL-Sub | | | PTBXL-Form | | | PTBXL-Rhythm | | | CPSC2018 | | | CSN | | |
|---|---|---|---|---|---|---|---|---|---|---|---|---|---|---|---|---|---|---|
| | 1% | 10% | 100% | 1% | 10% | 100% | 1% | 10% | 100% | 1% | 10% | 100% | 1% | 10% | 100% | 1% | 10% | 100% |
| Random Init | 70.45 | 77.09 | 81.61 | 55.82 | 67.60 | 77.91 | 55.82 | 62.54 | 73.00 | 46.26 | 62.36 | 79.29 | 54.96 | 71.47 | 78.33 | 47.22 | 63.17 | 73.13 |
| SimCLR | 63.41 | 69.77 | 73.53 | 60.84 | 68.27 | 73.39 | 54.98 | 56.97 | 62.52 | 51.41 | 69.44 | 77.73 | 59.78 | 68.52 | 76.54 | 59.02 | 67.26 | 73.20 |
| BYOL | 71.70 | 73.83 | 76.45 | 57.16 | 67.44 | 71.64 | 48.73 | 61.63 | 70.82 | 41.99 | 74.40 | 77.17 | 60.88 | 74.42 | 78.75 | 54.20 | 71.92 | 74.69 |
| BarlowTwins | 72.87 | 75.96 | 78.41 | 62.57 | 70.84 | 74.34 | 52.12 | 60.39 | 66.14 | 50.12 | 73.54 | 77.62 | 55.12 | 72.75 | 78.39 | 60.72 | 71.64 | 77.43 |
| MoCo-v3 | 73.19 | 76.65 | 78.26 | 55.88 | 69.21 | 76.69 | 50.32 | 63.71 | 71.31 | 51.38 | 71.66 | 74.33 | 62.13 | 76.74 | 77.68 | 54.61 | 74.26 | 77.68 |
| SimSiam | 73.15 | 72.70 | 75.63 | 62.52 | 69.31 | 76.38 | 55.16 | 62.91 | 71.31 | 49.30 | 69.47 | 75.92 | 58.35 | 72.89 | 75.31 | 58.25 | 68.61 | 77.41 |
| TS-TCC | 70.73 | 75.88 | 78.91 | 53.54 | 66.98 | 77.87 | 48.04 | 61.79 | 71.18 | 43.34 | 69.48 | 78.23 | 57.07 | 73.62 | 78.72 | 55.26 | 68.48 | 76.79 |
| CLOCS | 68.94 | 73.36 | 76.31 | 57.94 | 72.55 | 76.24 | 51.97 | 57.96 | 72.65 | 47.19 | 71.88 | 76.31 | 59.59 | 77.78 | 77.49 | 54.38 | 71.93 | 76.13 |
| ASTCL | 72.51 | 77.31 | 81.02 | 61.86 | 68.77 | 76.51 | 44.14 | 60.93 | 66.99 | 52.38 | 71.98 | 76.05 | 57.90 | 77.01 | 79.51 | 56.40 | 70.87 | 75.79 |
| CRT | 69.68 | 78.24 | 77.24 | 61.98 | 70.82 | 78.67 | 46.41 | 59.49 | 68.73 | 47.44 | 73.52 | 74.41 | 58.01 | 76.43 | 82.03 | 56.21 | 73.70 | 78.80 |
| ST-MEM | 61.12 | 66.87 | 71.36 | 54.12 | 57.86 | 63.59 | 55.71 | 59.99 | 66.07 | 51.12 | 65.44 | 74.85 | 56.69 | 63.32 | 70.39 | 59.77 | 66.87 | 71.36 |
| MELP | 85.82 | 87.61 | 87.87 | 79.22 | 84.40 | 87.46 | 63.41 | 76.71 | 83.30 | 88.83 | 94.65 | 96.91 | 88.54 | 91.75 | 94.32 | 78.25 | 84.83 | 90.17 |
| D-BETA | 83.15 | 88.36 | 90.11 | 77.74 | 82.92 | 85.15 | 70.10 | 78.91 | 83.98 | 86.61 | 92.83 | 96.71 | 85.46 | 91.35 | 94.92 | 80.04 | 87.36 | 90.71 |
| MERL | 82.39 | 86.27 | 88.67 | 64.90 | 80.56 | 84.72 | 58.26 | 72.43 | 79.65 | 53.33 | 82.88 | 88.34 | 70.33 | 85.32 | 90.57 | **66.60** | **82.74** | 87.95 |
| CLEP | 84.73 | 89.45 | 90.24 | 69.61 | **86.39** | 92.86 | 68.64 | 73.23 | 83.27 | 62.39 | **92.80** | 90.81 | 83.79 | 94.02 | 97.22 | 63.54 | 80.76 | 94.01 |
| FG-CLEP | **85.49** | **90.34** | 91.33 | **70.86** | 86.46 | **93.36** | **69.53** | **75.53** | **86.26** | **69.61** | 92.11 | **94.64** | **84.08** | **94.33** | **97.42** | 63.32 | 80.01 | **94.17** |

x', where x represents any of the 12 leads). This further indicates FG-CLEP's fine-grained waveform feature capture capability. However, the ensemble inference method often proves detrimental to CLEP, as seen in PTBXL-Super, PTBXL-Form, and CPSC2018.

Additionally, we applied our generated fine-grained reports to other methods, METS (Li et al., 2024) and MERL (Liu et al., 2024b), to validate the generalizability. The results demonstrate that our fine-grained reports can also enhance the performance of these methods.

## 4.4 LINEAR EVALUATION

We aim to evaluate the learned model transferability to downstream supervised tasks. We froze the ECG encoder and fine-tuned a randomly initialized linear classification head on the training data with binary cross-entropy loss. We compared a series of contrastive and generative self-supervised learning methods. Results in Table 2 show that FG-CLEP still achieves the best performances across all methods (Chen et al., 2020; Grill et al., 2020; Zbontar et al., 2021; Chen et al., 2021; Chen & He, 2021; Eldele et al., 2021; Kiyasseh et al., 2021; Wang et al., 2023; Zhang et al., 2023; Na et al., 2024; Liu et al., 2024b) in most scenarios.

Furthermore, when comparing the linear probe result in Table 2 with the zero-shot result in Table 1, we surprisingly find that FG-CLEP's zero-shot predictions are comparable to Linear Probe evaluations using 10% of the data in PTBXL-Sub, PTBXL-Form, CPSC2018, and CSN. Additionally, the zero-shot performance in PTBXL-Form is comparable to the full 100% Linear Probe evaluation. This further confirms the robustness and generalizability of our framework.

## 5 ANALYSIS

In this section, we conduct a series of experiments to provide an in-depth analysis of FG-CLEP. The reported metrics reflect the average zero-shot AUC across the six datasets described above.

### 5.1 ABLATION STUDY

To evaluate the effectiveness of our proposed fine-grained alignment with tag-specific ECG segments, the coarse-to-fine training with fine-grained reports, and the false negative mitigation loss function, we conducted a series of ablation studies. The results are presented in Table 3.

Table 3: Results of ablation study.

| Model Setting | AUC |
|---|---|
| FG-CLEP(default) | 83.03 |
| w/o Fine-Grained Alignment | 82.27 |
| w/o Fine-Grained Report | 80.78 |
| CLEP + 3 epochs (original reports) | 80.03 |
| FG-CLEP (trained from scratch) | 83.01 |
| w/o False Negative Mitigation | 81.67 |

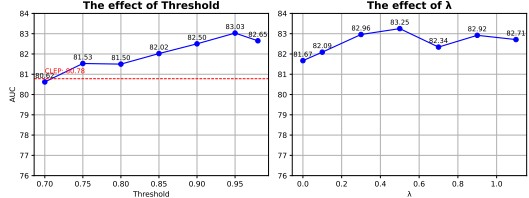

Figure 3: Effect of Threshold and $\lambda$.

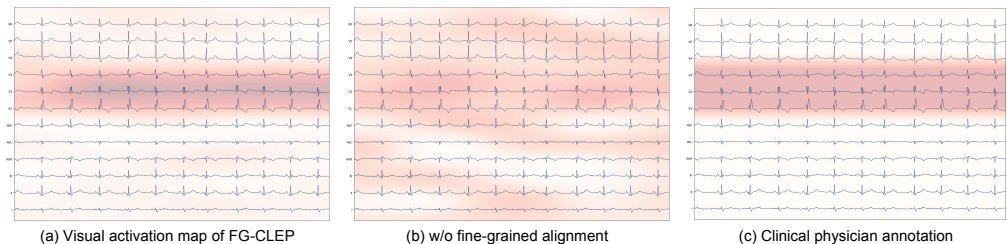

(a) Visual activation map of FG-CLEP    (b) w/o fine-grained alignment    (c) Clinical physician annotation

Figure 4: Visual activation maps generated by FG-CLEP and by the model without fine-grained alignment, using the text prompt *'RSR' Pattern in V1-V3'*.

Specifically, when removing the fine-grained alignment and performing contrastive learning only at the whole-report and ECG level, the performance drops significantly. We also compared FG-CLEP with CLEP trained without the additional 3 epochs of fine-grained report training. The results demonstrate that FG-CLEP significantly outperforms CLEP. To further confirm that the performance gains stem from the fine-grained reports rather than additional training epochs, we trained CLEP for 3 extra epochs using the original reports. The findings indicate that CLEP converges within 10 epochs, and the additional training even risks overfitting, leading to a slight performance drop. This further validates the importance of fine-grained reports. We also tested training FG-CLEP from scratch using fine-grained reports instead of continuing from CLEP. The results show no performance improvement but increased computational costs, supporting our default approach of continuing training from CLEP. Additionally, to verify the effectiveness of mitigating false negatives, we evaluated the performance without $L_{\text{fnm}}$. The results reveal a significant performance drop, highlighting the efficacy of our proposed loss function in addressing false negatives.

Finally, to assess the robustness of our method, we conducted ablation experiments on the loss hyperparameter $\lambda$ and the threshold for selecting fine-grained waveform features. As shown in Figure 3, our model maintains strong performance across different $\lambda$ and threshold values, demonstrating its robustness. To ensure high precision in generating waveform features, we set a relatively high default threshold of 0.95.

## 5.2 VISUAL ACTIVATION MAP

To further demonstrate the fine-grained alignment between report tags and specific ECG segments, we present visual activation maps generated by FG-CLEP and by the model without fine-grained alignment using the text prompt 'RSR' Pattern in V1-V3', which is a key waveform feature for diagnosing right bundle branch block. As shown in Figure 4, FG-CLEP effectively captures ECG segments relevant to the text prompt, whereas the model without fine-grained alignment fails to accurately localize the specific ECG segments. This further validates the effectiveness of our approach in achieving fine-grained alignment.

## 5.3 VALIDATION OF FINE-GRAINED REPORT

In our training dataset MIMIC-ECG, nearly 50% of reports do not mention any waveform features. As mentioned earlier, rather than directly using the waveform features generated by the LLM as ground truth, we validate these features with CLEP to produce fine-grained reports. To evaluate the accuracy and reliability of the generated fine-grained reports, as well as the effectiveness of CLEP for validation, we randomly selected 100 ECGs that lacked waveform features. Three medical students independently annotated these ECGs for five key waveform features. The final labels

Table 4: AUC of Generated Fine-Grained Reports (with/without verification).

| Waveform Feature | w/o Veri. | w Veri. |
|---|---|---|
| Non-specific ST elevation | 71.42 | 83.42 |
| Long QT-interval | 84.25 | 93.25 |
| Abnormal QRS | 76.68 | 91.68 |
| Prolonged PR interval | 75.13 | 86.13 |
| Inverted T-waves | 74.79 | 88.79 |

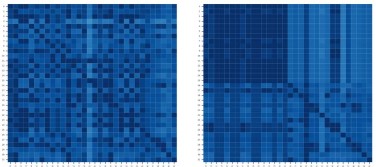

Figure 5: **The Heatmap of Semantic Similarity Matrix.** Left: a random batch; Right: with the first 16 as normal ECG and the last 16 as abnormal ECG.

were determined by majority vote, and the accuracy of the generated fine-grained reports was calculated accordingly. Since the same waveform feature may have different (granular) expressions, we computed the similarity between the generated reports and the five waveform features to obtain AUC values. As shown in Table 4, the AUC of waveform features directly generated by the LLM is relatively low, which aligns with expectations due to the hallucination issues of LLMs and the non-bijective relationship between ECG waveform features and diagnoses. However, after validation with CLEP, the AUC of the fine-grained reports improves significantly, demonstrating the effectiveness and reliability of the coarse-to-fine training process in generating fine-grained reports.

## 5.4 SEMANTIC SIMILARITY MATRIX

We visualize the semantic similarity matrix in Figure 5. The left side shows the semantic similarity matrix from a random batch. As illustrated, ECGs and tags from different records may share similarities to some extent. Ignoring these similarities would result in a diagonal matrix with ones on the diagonal and zeros elsewhere, which is obviously wrong. The right side displays a semantic similarity matrix where the first 16 entries are normal ECGs and the last 16 are abnormal ECGs. The matrix effectively captures the semantic similarities of the normal ECGs.

## 5.5 DIFFERENT COMPONENTS

We conducted experiments to evaluate the performance of our framework using different LLMs and text encoders. The results are presented in Table 5. The findings indicate that our framework is robust across various components. Specifically, for different LLMs (Abdin et al., 2024; Jiang et al., 2023a; AI@Meta, 2024; Labrak et al., 2024; Ankit Pal, 2024), larger LLM provide some performance improvements, though the gains are not substantial. While domain-specific LLMs possess more medical knowledge, our method requires formatting the waveform feature outputs, an area where domain-specific models are less effective, resulting in performance that does not surpass general-purpose models. For different text encoders (Alsentzer et al., 2019; Gu et al., 2021; Jin et al., 2023; Lee et al., 2020), our framework consistently achieves significant improvements.

Table 5: Results on Different LLM/Text Encoder/ECG Encoder.

| Model Type | CLEP | FG-CLEP ↑ |
|---|---|---|
| **Different LLMs** | | |
| LLaMA3-8B-Instruct | 80.78 | 83.03 |
| LLaMA3-70B-Instruct | 80.78 | 83.42 |
| Qwen3-8B | 80.78 | 83.00 |
| Phi-3-mini-4k-instruct | 80.78 | 81.51 |
| Mistral-7B-Instruct-v0.2 | 80.78 | 82.12 |
| BioMistral-7B | 80.78 | 82.67 |
| LLaMA3-OpenBioLLM-8B | 80.78 | 83.23 |
| **Different Text Encoders** | | |
| BioClinicalBERT | 80.78 | 83.03 |
| PubMedBERT | 80.92 | 82.07 |
| Med-CPT | 78.55 | 81.21 |
| BioBERT | 78.63 | 81.20 |

## 6 CONCLUSION

In this paper, we introduced FG-CLEP, a fine-grained ECG-text contrastive learning framework that enhances waveform understanding by aligning specific ECG segments with report tags. Our coarse-to-fine training process leverages large language models to recover missing waveform features and incorporates a semantic similarity matrix to mitigate false negatives. Extensive experiments on six datasets demonstrate that FG-CLEP achieves state-of-the-art performance in both zero-shot and linear probing settings. These results highlight the effectiveness and generalizability of FG-CLEP.

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

## A    DATASET ANALYSIS

We pre-train the FG-CLEP using the MIMIC-ECG dataset and test it on the PTB-XL, CPSC2018, and CSN datasets. All the ECGs in the datasets are 12-lead recordings. The PTB-XL dataset can be further divided into four subsets, and we follow the official train:validation:test split. For CPSC2018 and CSN, we split the dataset as 70%:10%:20% for the train:validation:test split. The statistics of the datasets used are presented in Table 6.

**MIMIC-ECG** The MIMIC-ECG dataset contains nearly 800,000 ECG-report pairs from approximately 160,000 unique patients. These diagnostic ECGs utilize 12 leads and are 10 seconds in duration, with a sampling rate of 500 Hz.

**PTB-XL** The PTB-XL ECG dataset is a large dataset of 21,799 clinical 12-lead ECGs from 18,869 patients of 10-second length. There are four subsets with multi-label classification tasks: Superclass (5 categories), Subclass (23 categories), Form (19 categories), and Rhythm (12 categories). Notably, these four subsets have different numbers of samples.

**CPSC2018** This publicly accessible dataset comprises 6,877 standard 12-lead ECG records, each sampled at a rate of 500 Hz, with durations ranging from 6 to 60 seconds. The dataset is annotated with 9 distinct labels.

**Chapman-Shaoxing-Ningbo (CSN)** This dataset contains 12-lead ECGs of 45,152 patients with a 500 Hz sampling rate. It features multiple common rhythms and additional cardiovascular conditions, all labeled by professional experts.

## B    PSEUDO CODE

The pseudo-code of our FG-CLEP training process is shown in algorithm 1

Table 6: The statistics of used datasets.

| Pretrain | # ECGs | # Reports | | |
|---|---|---|---|---|
| MIMIC-ECG | 773,268 | 773,268 | | |
| **Evaluation** | **# Train** | **# Valid** | **# Test** | **# Classes** |
| PTB-XL Super | 17,084 | 2,146 | 2,158 | 5 |
| PTB-XL Sub | 17,084 | 2,146 | 2,158 | 23 |
| PTB-XL Form | 7,197 | 901 | 880 | 19 |
| PTB-XL Rhythm | 16,832 | 2,100 | 2,098 | 12 |
| CPSC2018 | 4,800 | 684 | 1,383 | 9 |
| CSN | 31,606 | 4,515 | 9,031 | 51 |

---

**Algorithm 1** FG-CLEP Training Process

---

1: **Input:** $D = \{(x_{\mathrm{ecg}_i}, x_{\mathrm{txt}_i}) \mid i \in [0, n)\}$
2: **Output:** FG-CLEP
3: Perform contrastive training on CLEP using $D$
4: Generate fine-grained reports
5: **for** $i = 0$ to $n - 1$ **do**
6:    $f_{\mathrm{features}} = \mathrm{LLM}(x_{\mathrm{txt}_i}, \mathrm{prompt})$
7:    **for** $j = 1$ to $m$ **do**
8:       where $m$ is the number of waveform features generated
9:       $\mathrm{sim} = \mathrm{CLEP}(x_{\mathrm{ecg}_i}, f_j)$
10:       **if** $\mathrm{sim} > \mathrm{threshold}$ **then**
11:          $x_{\mathrm{txt}_i} = x_{\mathrm{txt}_i} + f_j$
12:       **end if**
13:    **end for**
14: **end for**
15: Continue training CLEP on $\{(x_{\mathrm{ecg}_i}, x_{\mathrm{txt}_i})\}$ to obtain FG-CLEP

---

## C   Running Cases for Generating Fine-Grained Reports

We present three case studies illustrating how the fine-grained reports with waveform features are generated step by step, as shown in Figure 6, 7, 8. The LLM is prompted with two explicit chain-of-thought instructions: **'What waveform features are most likely to be present in electrocardiograms with these symptoms?'** followed by **'Organize these waveform features into a Python list, with each item representing a distinct waveform feature.'**

## D   ECG-Text Retrieval

We attempted to use FG-CLEP to retrieve electrocardiograms (ECGs) from the MIMIC-ECG dataset (Gow et al.) through text. To test our model's ability to capture fine-grained waveform features, we tested a series of typical waveform features such as 'RSR' Pattern, 'Inverted T-waves,' and 'Low QRS voltages.' Figure 9 shows the Top 3 retrieved ECGs with probabilities all greater than 0.99. Our model demonstrated strong capability in retrieving ECGs through waveform feature text, which can lead to two applications: (1) Helping doctors quickly find similar cases or specific ECG patterns, aiding in diagnosis and treatment decision-making; (2) In medical education, text-based retrieval can quickly find typical ECG cases, assisting in teaching and training, thereby improving educational effectiveness.

**ID: p14578883/s43946433/43946433**
**Report**: Atrial fibrillation., Right bundle branch block, Abnormal ECG
**LLM Output:**
Here is the list of waveform features organized into a Python list:

```
waveform_features = [
    "Irregularly irregular rhythm",
    "Absent or fragmented P-waves",
    "Wide QRS complexes (>120 ms)",
    "Variable RR intervals",
    "Fibrillatory waves (f-waves) or oscillations",
    "Wavy or sawtooth pattern",
    "Notched or slurred R wave in leads I, II, and V5-V6",
    "Inverted or absent R wave in lead V1",
    "Delta wave (slurred upstroke of the QRS complex) in leads I, II, and V5-V6",
    "RSR' pattern in leads V1-V3",
    "ST segment changes (elevation, depression, or inversion)",
    "T wave changes (inversion, flattening, or peaking)",
    "QT interval prolongation or shortening",
    "PR interval prolongation or shortening",
    "U waves or other abnormal waveforms"
]
```

Note that this list is not exhaustive, and there may be other waveform features that are not included here. Additionally, the presence of these features does not necessarily confirm a specific diagnosis, and further evaluation and testing may be necessary to establish a definitive diagnosis.
**Potential Waveform features:**
['Irregularly irregular rhythm', 'Absent or fragmented P-waves', 'Wide QRS complexes (>120 ms)', 'Variable RR intervals', 'Fibrillatory waves (f-waves) or oscillations', 'Wavy or sawtooth pattern', 'Notched or slurred R wave in leads I, II, and V5-V6', 'Inverted or absent R wave in lead V1', 'Delta wave (slurred upstroke of the QRS complex) in leads I, II, and V5-V6', "RSR' pattern in leads V1-V3", 'ST segment changes (elevation, depression, or inversion)', 'T wave changes (inversion, flattening, or peaking)', 'QT interval prolongation or shortening', 'PR interval prolongation or shortening', 'U waves or other abnormal waveforms']
**Probility:**
[0.985297, 0.49500474, 0.95943975, 0.9952996, 0.96285915, 0.84171605, 0.88279337, 0.9455562, 0.98828167, 0.99366695, 0.62816966, 0.9148475, 0.99069655, 0.049276203, 0.9565827]
**Fine-Grained Report**: Atrial fibrillation., Right bundle branch block, Abnormal ECG, Irregularly irregular rhythm,Wide QRS complexes (>120 ms),Variable RR intervals,Fibrillatory waves (f-waves) or oscillations,Delta wave (slurred upstroke of the QRS complex) in leads I, II, and V5-V6,RSR' pattern in leads V1-V3,QT interval prolongation or shortening,U waves or other abnormal waveforms

Figure 6: Running Case1 with Atrial fibrillation and Right bundle branch block.

**ID: p11799366/s45322712/45322712**

**Report**: Probable atrial fibrillation, ventricular couplets, Long QTc interval, Ant/septal+lateral ST-T changes suggest myocardial infarction, Repolarization changes may be partly due to rhythm, Abnormal ECG

**LLM Output:**

Here is the list of waveform features organized in a Python list:

```
waveform_features = [
    "Irregularly irregular rhythm",
    "Absent or fragmented P waves",
    "Fibrillatory waves (f-waves) or oscillations in the atrial region",
    "Premature ventricular contractions (PVCs) with a different morphology from the normal QRS complex",
    "Ventricular couplets: two consecutive PVCs with a normal QRS complex in between",
    "Long QTc interval (> 440 ms for men, > 460 ms for women)",
    "Abnormal T-wave morphology (e.g., 'saddle-shaped' or 'inverted' T-wave)",
    "ST-segment elevation or depression in anteroseptal and lateral leads (e.g., V1-V3, I, aVL)",
    "T-wave inversion or abnormal T-wave morphology in anteroseptal and lateral leads",
    "Abnormal QRS complex morphology (e.g., wide or bizarre QRS complexes)",
    "Prolonged PR interval",
    "Abnormal QRS axis"
]
```

Note that I've kept the descriptions concise and focused on the key features, as they would be in a clinical setting.

**Potential Waveform features:**
['Irregularly irregular rhythm', 'Absent or fragmented P waves', 'Fibrillatory waves (f-waves) or oscillations in the atrial region', 'Premature ventricular contractions (PVCs) with a different morphology from the normal QRS complex', 'Ventricular couplets: two consecutive PVCs with a normal QRS complex in between', 'Long QTc interval (> 440 ms for men, > 460 ms for women)', "Abnormal T-wave morphology (e.g., 'saddle-shaped' or 'inverted' T-wave)", 'ST-segment elevation or depression in anteroseptal and lateral leads (e.g., V1-V3, I, aVL)', 'T-wave inversion or abnormal T-wave morphology in anteroseptal and lateral leads', 'Abnormal QRS complex morphology (e.g., wide or bizarre QRS complexes)', 'Prolonged PR interval', 'Abnormal QRS axis']

**Probility:**
[0.9616148, 0.9832206, 0.9936021, 0.9954572, 0.9854593, 0.99531144, 0.9919208, 0.9786107, 0.99453413, 0.99271095, 0.27789757, 0.51983744]

**Fine-Grained Report**: Probable atrial fibrillation, ventricular couplets, Long QTc interval, Ant/septal+lateral ST-T changes suggest myocardial infarction, Repolarization changes may be partly due to rhythm, Abnormal ECG, Irregularly irregular rhythm,Absent or fragmented P waves,Fibrillatory waves (f-waves) or oscillations in the atrial region,Premature ventricular contractions (PVCs) with a different morphology from the normal QRS complex,Ventricular couplets: two consecutive PVCs with a normal QRS complex in between,Long QTc interval (> 440 ms for men, > 460 ms for women),Abnormal T-wave morphology (e.g., 'saddle-shaped' or 'inverted' T-wave),ST-segment elevation or depression in anteroseptal and lateral leads (e.g., V1-V3, I, aVL),T-wave inversion or abnormal T-wave morphology in anteroseptal and lateral leads,Abnormal QRS complex morphology (e.g., wide or bizarre QRS complexes)

Figure 7: Running Case2 with Myocardial Infarction.

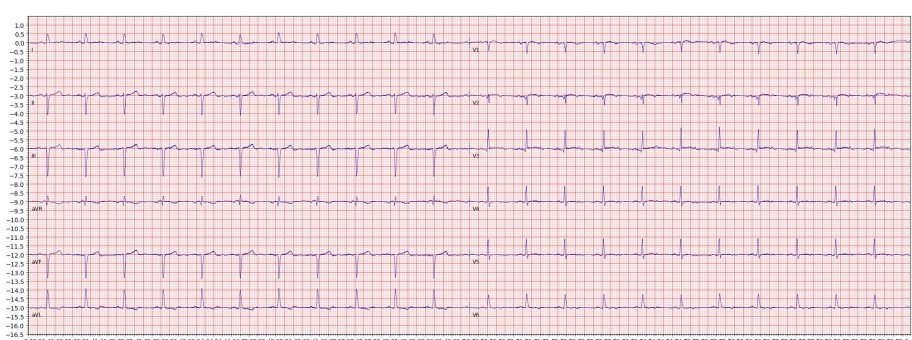

**ID: p10474653/s42221714/42221714**

**Report**: Sinus rhythm, Left anterior fascicular block, Possible septal infarct - age undetermined, Left ventricular hypertrophy, Lateral T wave changes are probably due to ventricular hypertrophy, Abnormal ECG

**LLM Output:**

Here is the list of waveform features organized in a Python list:

```
waveform_features = [
    "Sinus rhythm",
    "Left axis deviation (LAD) >30 ˚ ",
    "Q waves in leads I, II, or V5-V6",
    "ST segment depression in leads I, II, or V5-V6",
    "T wave inversion in leads I, II, or V5-V6",
    "Increased amplitude of the QRS complex (>3.5 mm) in leads I, II, or V5-V6",
    "Increased duration of the QRS complex (>120 ms)",
    "Deep S waves in leads I, II, or V5-V6",
    "T wave inversion or flattening in leads I, aVL, V5, and V6",
    "ST segment elevation or depression in other leads",
    "Prolonged QT interval",
    "Abnormal P wave morphology or axis"
]
```

Note that I've tried to be as specific as possible when describing each waveform feature, but some features may overlap or be related to each other.

**Potential Waveform features:**

['Sinus rhythm', 'Left axis deviation (LAD) >30 ˚ ', 'Q waves in leads I, II, or V5-V6', 'ST segment depression in leads I, II, or V5-V6', 'T wave inversion in leads I, II, or V5-V6', 'Increased amplitude of the QRS complex (>3.5 mm) in leads I, II, or V5-V6', 'Increased duration of the QRS complex (>120 ms)', 'Deep S waves in leads I, II, or V5-V6', 'T wave inversion or flattening in leads I, aVL, V5, and V6', 'ST segment elevation or depression in other leads', 'Prolonged QT interval', 'Abnormal P wave morphology or axis']

**Probility:**

[0.9861351, 0.5359271, 0.33153382, 0.71545005, 0.49900547, 0.42786044, 0.61200434, 0.18875751, 0.8920379, 0.78382677, 0.88176095]

**Fine-Grained Report**: Sinus rhythm, Left anterior fascicular block, Possible septal infarct - age undetermined, Left ventricular hypertrophy, Lateral T wave changes are probably due to ventricular hypertrophy, Abnormal ECG, Left axis deviation (LAD) >30 ˚

Figure 8: Running Case3 with Hypertrophy.

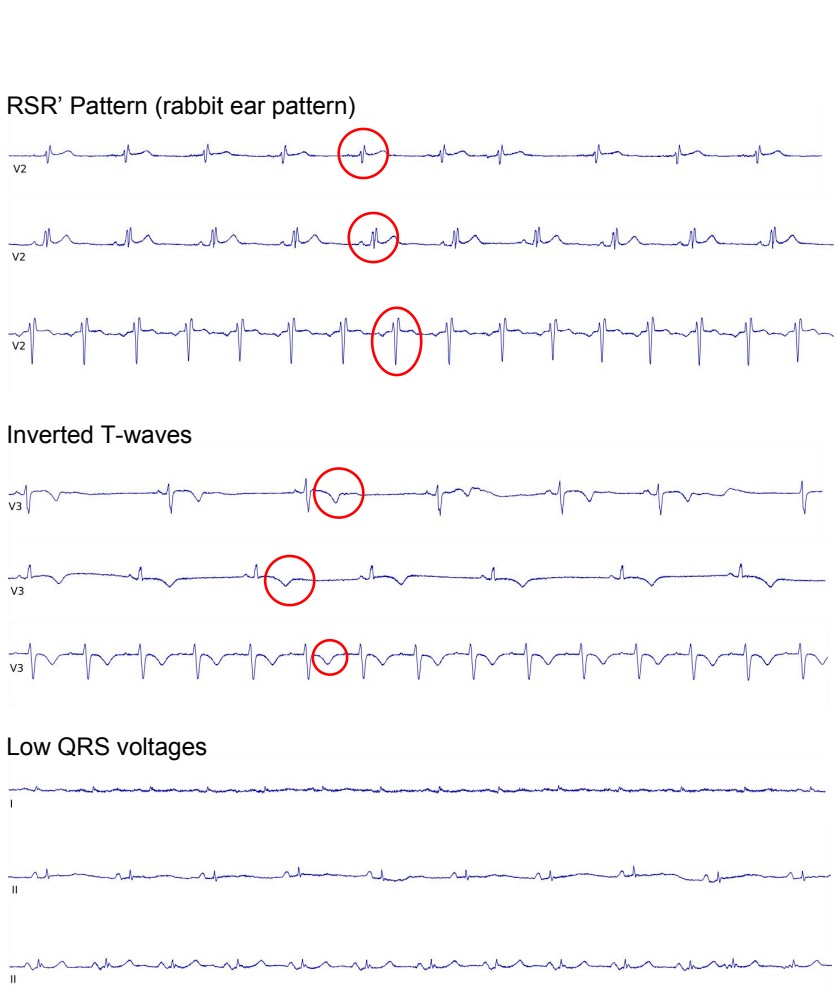

Figure 9: Top 3 retrieved ECG using FG-CLEP.

