# OpenReview forum: "Fine-grained Contrastive Learning for ECG-Report Alignment with Waveform Enhancement"
_ICLR.cc/2026/Conference — Submitted to ICLR 2026_

### Official Review · Reviewer_YzBY · 2025-10-24

**Soundness:** 2
**Presentation:** 3
**Contribution:** 2
**Rating:** 2
**Confidence:** 5

**Summary:**

This paper presents FG-CLEP, a fine-grained ECG–report alignment model that links ECG patches with report tags.

**Strengths:**

- Introduces a cross-attention patch-tag alignment and LLM-based fine-grained training pipeline for ECG-text pretraining.

**Weaknesses:**

- This work has limited technical novelty. The work depends on the LLM usage instead of core modeling. Also, the LLM-generated GF reports are clinically unverifiable and risk introducing noise or bias and especially even label leakage when doing zero-shot experiments.

- Lack comparisions with recent ECG-text modeling works.

**Questions:**

- How to ensure ECG patches with specific patch length align to any of clinical report tags?

- Please compare the work with recent advances on ECG-Text pretraining works [1,2]. In many evaluation setting, the performance are poorer than them, whether the work uses “FG-” or not.

- Furthermore, [2] also point out the close problem with false negative samples (report level). Please provide additional experiments to compare N3S in [2] and the FNM in this work.

- Please add linear probing experiment with METS model and compare with the work.

- In ablation study, w/o Fine-Grained Alignment performance is very close (<1%) with the proposed model (default) while the w/o Fine-Grained Report performance is clearer. Does this show that the work highly rely on additional LLM usage and used trick like ensemble?

- Furthermore, looking at linear probing results (table 2), where only ECG encoder is used, the gap between with “FG-“ and without “FG-” is minor. Does this mean the “FG-” just slightly boost the ECG encoder (which I suppose, is the core component in real-world deployment).

- Figure 4 is not mentioned in any text.

- It seems to be unable to access to the code provided.

[1] Wang, Fuying, Jiacheng Xu, and Lequan Yu. "From Token to Rhythm: A Multi-Scale Approach for ECG-Language Pretraining." Forty-second International Conference on Machine Learning.

[2] Hung, Manh Pham, Aaqib Saeed, and Dong Ma. "Boosting Masked ECG-Text Auto-Encoders as Discriminative Learners." Forty-second International Conference on Machine Learning.

---

> ### Author Response · Authors · 2025-11-18
>
> Thank you very much for your thoughtful feedback and constructive comments. We truly appreciate the opportunity to clarify key aspects of our work. We kindly request that you reconsider the score in light of the clarifications below.
>
> **Weakness1: This work has limited technical novelty. The work depends on the LLM usage instead of core modeling.**
>
> Using LLMs to generate waveform features is only one component of our contribution, and we explicitly emphasize that directly using raw LLM outputs is not reliable. To address this, we introduce a coarse-to-fine training pipeline in which the CLEP model validates and filters LLM-generated candidates before incorporating them into training. In addition, we design a fine-grained alignment architecture and propose a dedicated false negative mitigation mechanism tailored for tag-specific alignment—both of which constitute core modeling innovations independent of LLM usage.
>
> **Weakness2: LLM-generated reports are clinically unverifiable and risk introducing noise or bias and especially even label leakage when doing zero-shot experiments.**
>
> Actually, we conducted human validation: three junior medical students independently annotated ECG waveform features for evaluation (Section 5.3), and the results are presented in Table 4. We did not provide any downstream task labels as prompts to the LLM, so there is no label leakage.
>
> **Weakness3: We have added the two comparisons suggested by the reviewer in the revised version, and our model outperforms both baselines in both zero-shot and linear-probe settings. Detailed in Q2 below**
>
> **Q1: How to ensure ECG patches with specific patch length align to any of clinical report tags?**
>
> We obtain tag-specific ECG representations through cross-attention to perform the alignment. The patch length does affect the precision: patches that are too coarse reduce accuracy, while patches that are too fine increase computational cost. As shown in Figure 4, our chosen patch length already provides sufficient precision.
>
> **Q2: Please compare the work with recent advances on ECG-Text pretraining works. In many evaluation setting, the performance are poorer than them, whether the work uses “FG-” or not.**
>
> We have added comparisons with these two works in the revised version. In fact, **our method performs better in most cases**. **kindly remind that in MELP, the order of the six datasets is different from that used in most other works, which may lead to misreading the columns.**
>
> | macro AUC | PTB-XL-Super | PTBXL-Sub | PTBXL-Form | PTBXL-Rhythm | CPSC2018 | CSN  |
> |-----------|--------------|-----------|------------|--------------|----------|------|
> | MELP      | 76.2         | 81.2      | 69.1       | 85.4         | 84.2     | 77.6 |
> | D-BETA    | 76.2         | 75.9      | 66.1       | 88.6         | 80.1     | 76.3 |
> | CLEP      | 78.0         | 82.4      | 68.0       | 89.5         | 86.0     | 80.9 |
> | FG-CLEP   | 80.1         | 84.5      | 68.5       | 93.0         | 88.9     | 83.2 |
>
>
> | Method   | PTBXL-Super 1% | PTBXL-Super 10% | PTBXL-Super 100% | PTBXL-Sub 1% | PTBXL-Sub 10% | PTBXL-Sub 100% | PTBXL-Form 1% | PTBXL-Form 10% | PTBXL-Form 100% | PTBXL-Rhythm 1% | PTBXL-Rhythm 10% | PTBXL-Rhythm 100% | CPSC2018 1% | CPSC2018 10% | CPSC2018 100% | CSN 1% | CSN 10% | CSN 100% |
> |----------|----------------|-----------------|------------------|--------------|----------------|-----------------|----------------|------------------|-------------------|-------------------|---------------------|----------------------|--------------|----------------|-----------------|---------|-----------|------------|
> | MELP     | 85.82 | 87.61 | 87.87 | 79.22 | 84.40 | 87.46 | 63.41 | 76.71 | 83.30 | 88.83 | 94.65 | 96.91 | 88.54 | 91.75 | 94.32 | 78.25 | 84.83 | 90.17 |
> | D-BETA   | 83.15 | 88.36 | 90.11 | 77.74 | 82.92 | 85.15 | 70.10 | 78.91 | 83.98 | 86.61 | 92.83 | 96.71 | 85.46 | 91.35 | 94.92 | 80.04 | 87.36 | 90.71 |
> | CLEP     | 84.73 | 89.45 | 90.24 | 69.61 | 86.39 | 92.86 | 68.64 | 73.23 | 83.27 | 62.39 | 92.80 | 90.81 | 83.79 | 94.02 | 97.22 | 63.54 | 80.76 | 94.01 |
> | FG-CLEP  | 85.49 | 90.34 | 91.33 | 70.86 | 86.46 | 93.36 | 69.53 | 75.53 | 86.26 | 69.61 | 92.11 | 94.64 | 84.08 | 94.33 | 97.42 | 63.32 | 80.01 | 94.17 |

---

> > ### Comment · Reviewer_YzBY · 2025-11-20
> >
> > Thanks the authors for the clarification and new experiments.
> >
> > Regarding the added comparison with the two references, I have two further concerns. First, in 1% setting in the linear probing table (i.e., with only ECG encoder), FG-CLEP performs significantly lower than the two baselines. This indicates that their ECG encoder itself is not strong. Second, but surprisingly, in the zero-shot table, i.e., when both text encoder and ECG encoder are presented, FG-CLEP achieves remarkably higher performance than the two baselines. This means the text encoder is the key for performance improvement. Looking at the pre-training step, the authors added some tags, which seems to be sort of of 'soft labels' in the training process. That is why I said there are potentially a data leakage issue (tag or text label leak) that attribute to the strong performance especially in zero-shot.

---

> ### Author Response · Authors · 2025-11-18
>
> **Q3: Furthermore, [2] also point out the close problem with false negative samples (report level). Please provide additional experiments to compare N3S in [2] and the FNM in this work.**
>
> First, N3S performs contrastive learning at the report level, selecting the report embedding that is farthest away as the negative sample. In contrast, our method conducts contrastive learning at the fine-grained tag level, which intensifies the false-negative problem, because different reports may still share common tags. Our semantic similarity matrix is designed to capture such false negatives—effectively informing the model that some tags from other reports should not be treated as negatives. N3S, however, manually filters negatives to ensure that all samples from other reports within a batch are true negatives. Although this strategy is less suitable for tag-level learning, we still conducted this additional experiment, as shown in Table 3 (zero-shot setting). This setting degrades FG-CLEP’s performance, but it still outperforms the model without any false-negative mitigation.
>
> | Model Setting                | AUC   |
> |------------------------------|-------|
> | FG-CLEP                      | 83.03 |
> | N3S                          | 82.10 |
> | w/o False Negative Mitigation| 81.67 |
>
> **Q4: Please add a linear-probing experiment with the METS model and compare it with the previous work.**
>
> METS is essentially the most basic and almost the first ECG-report contrastive learning model. It has since been improved by many subsequent works such as MERL and the two papers you mentioned. We already compared the two stronger methods you proposed in Q2. Moreover, METS is trained only on PTB-XL, whereas most later works are trained on the much larger MIMIC-ECG dataset.
>
> **Q5: In ablation study, w/o Fine-Grained Alignment performance is very close (<1%) with the proposed model (default) while the w/o Fine-Grained Report performance is clearer. Does this show that the work highly rely on additional LLM usage and used trick like ensemble?**
>
> We do use an LLM to assist in generating waveform features, **but simply using an LLM is not sufficient**. **As shown in Table 4, the raw outputs from the LLM are not reliable, mainly due to the non-bijective relationship between waveform features and diseases as well as hallucination issues.** We therefore designed a complete coarse-to-fine training pipeline, where CLEP is used to verify the LLM-generated attributes, and a carefully selected threshold (as shown in Figure 3) is applied to obtain high-quality fine-grained reports. It is also worth noting that these fine-grained reports lead to clear improvements when applied to other baselines as well (Table 1).
>
> **Q6: Furthermore, looking at linear probing results (table 2), where only ECG encoder is used, the gap between with “FG-“ and without “FG-” is minor. Does this mean the “FG-” just slightly boost the ECG encoder (which I suppose, is the core component in real-world deployment).**
>
> + First, **in real-world deployment we would prefer zero-shot transfer, which is actually the original intention of these CLIP-like approaches.**
>
> + Second, I don’t think we can conclude that “FG-” just slightly boosts the ECG encoder. The linear-probe setting involves fine-tuning on the downstream task, which actually weakens the visibility of the pretraining effect. If we want to assess the intrinsic quality of the ECG encoder, we should look directly at the zero-shot results. Also, more detailed ablation experiments can be found in Table 3, which provides much more comprehensive evidence than Table 2.
>
> **Q7: Figure 4 is not mentioned in any text.**
>
> Typo: the Figure 5 referenced in Section 5.2 should be corrected to Figure 4. Thanks.
>
> **Q8: It seems to be unable to access the code provided.**
>
> The code was prepared when the paper was submitted, and it has not expired since then. I just checked and it is accessible without any issues. You may try again — it might have been a network problem.

---

> ### Author Response · Authors · 2025-11-20
>
> Thank you for your response.
>
> + First, regardless of whether it is zero-shot or linear-probe, both settings can actually reflect the capability of the ECG encoder. In fact, I believe that fine-tuning on downstream task data may weaken the degree to which the ECG encoder’s inherent capability is manifested.
>
> + Regarding the 1% linear probing scenario where our method performs lower than MELP and D-BETA, especially on the Rhythm task (FG-CLEP 69.61 vs. MELP 88.83 vs. D-BETA 86.61), we also find this surprising, because **our results are close to all other baselines in Table 2, while these two methods are unusually high**. One noteworthy point is that our FG-CLEP achieves 93.02 on Rhythm zero-shot, but drops to 69.61 with 1% linear probe. **In fact, linear probing with too little data can perform worse than zero-shot. A similar phenomenon is also observed in the well-known MERL model: zero-shot 78.50 vs. 1% linear probe 53.33 on Rhythm.** In contrast, **the two methods you mentioned show 1% linear probe performance that is close to or even exceeds their zero-shot performance**. We further found that both methods performed linear probing with an extremely large number of epochs; for example, D-BETA used 200 epochs for linear probe, whereas we used only 30 epochs. We suspect this may be the reason. However, using too many epochs comes with a high risk of overfitting.
>
> + As for your comment that the text encoder is the key factor behind the performance improvement, we actually conducted experiments with different text encoders (Table 5), and the results do not show a significant difference.
>
> + Regarding the concern of data leakage: the prompt we used in the LLM was **"What waveform features are most likely to be present in ECG with these symptoms?"** We did not feed downstream task labels into the LLM through the prompt. Therefore, **this should be viewed more as a form of *data augmentation* rather than *data leakage*.**

---

> > ### Comment · Reviewer_YzBY · 2025-11-25
> >
> > While both zero-shot and linear-probe can reflect the capability of ECG encoder, the deviated performance gap with baselines indeed reveals the instability of the ECG encoder, which I believe is the key for ECG analysis. Some arguments in the response also cannot convince me, but I would like to increase the score slightly.

---

> ### Author Response · Authors · 2025-11-27
>
> Thank you for your patience and for your follow-up response.
>
> + Regarding the gap observed in the PTB-XL Rhythm, 1% linear-probe setting, we conducted substantial further analysis. We also increased the linear-probe training epochs from 30 to 200, but this brought no performance gain and instead resulted in overfitting trend. **This is expected because the 1% Rhythm split contains only 168 labeled samples, and training for 200 epochs is excessively large for such a small dataset.**
>
> + In fact, **our method matches and exceeds most baselines on PTB-XL Rhythm under the 1% linear-probe setting.** The exceptionally high performance of MELP (88.83) on Rhythm–1% stands out sharply compared with FG-CLEP (69.61), CLEP (62.39), and many other methods such as MERL (who introduced this benchmark, 53.33) and HeartLang (ICLR 2025) (62.08). **This unusually strong result on Rhythm is largely due to MELP’s Rhythm-specific architectural design, which is a core innovation of MELP. However, such architectural specialization is orthogonal to the contributions of our work, and their high Rhythm score should not be interpreted as evidence against the effectiveness of our method.**
>
> + **Except for this isolated outlier on Rhythm–1%, our method shows consistently strong performance across both zero-shot and linear-probe evaluations, demonstrating the overall effectiveness of our approach.**
>
> Thanks again for your response.

---

### Official Review · Reviewer_4kU3 · 2025-10-30

**Soundness:** 3
**Presentation:** 3
**Contribution:** 2
**Rating:** 6
**Confidence:** 4

**Summary:**

The paper presents FG-CLEP, an innovative framework that advances ECG–text alignment by introducing fine-grained contrastive learning between ECG patches and report tags.

**Strengths:**

The paper presents FG-CLEP, an innovative framework that advances ECG–text alignment by introducing fine-grained contrastive learning between ECG patches and report tags. Its coarse-to-fine training pipeline, which integrates large language models (LLMs) for recovering missing waveform features, demonstrates strong methodological creativity and practical relevance. Extensive experiments across six datasets show consistent performance gains in both zero-shot and linear probing tasks, validating the model’s robustness and generalizability. Visualizations (activation maps and retrieval results) provide qualitative support for fine-grained alignment.

**Weaknesses:**

The dependence on LLMs for generating waveform features may introduce bias or inconsistency, even with CLEP-based validation.
The evaluation lacks human expert assessment of fine-grained alignment quality beyond AUC metrics, limiting interpretability claims.
The training efficiency and computational cost of multi-stage fine-tuning and LLM querying are not clearly quantified, raising concerns about scalability in clinical deployment.
The model’s reliance on tag-level alignment assumes structured report formats, which may not generalize across healthcare systems or languages.
While the paper claims improvements in zero-shot settings, statistical significance testing is not reported.
There are several writing issues, i.e., the legend of Fig.2 is incomplete (line 230); ref information is incomplete (line 286).

**Questions:**

How does FG-CLEP perform on free-text clinical reports rather than tag-based structured ones?

What are the computational and time costs of the coarse-to-fine training pipeline when scaling to larger datasets?

Have you evaluated the clinical interpretability of tag-specific ECG activations with cardiologists?

How sensitive is FG-CLEP to LLM hallucinations or errors during waveform feature generation, especially when the validation threshold varies?

---

> ### Author Response · Authors · 2025-11-18
>
> **Weaknesses1: Potential Bias or Inconsistency Introduced by LLM-Generated Waveform Features**
>
> As discussed in the paper, raw LLM-generated waveform features are not directly trusted due to the non-bijective relationship between waveform patterns and diagnoses and the possibility of hallucinations. Therefore, we introduce CLEP-based validation to filter these candidates using the ECG signal itself, which substantially improves feature accuracy (Table 4). Regarding potential LLM-induced bias, Table 5 shows that FG-CLEP achieves consistent improvements across diverse LLMs, indicating that our coarse-to-fine pipeline is robust and does not rely on any particular LLM behavior.
>
> **Weakness2: Concern about Human Expert Assessment for Fine-Grained Alignment**
>
> In fact, the evaluation in Table 4 is based on human annotations. As mentioned in Section 5.3, we randomly selected 100 ECGs that lacked waveform features, and three medical students independently annotated five key waveform features for each ECG.
>
> **Weakness3: Concerns About Clinical Deployment Cost**
>
> The multi-stage fine-tuning only adds 3 additional epochs on top of the original 10 epochs, resulting in a modest increase in training cost. Although the LLM-based waveform enhancement introduces extra computation, all LLM queries are performed offline during training. The actual inference cost for clinical deployment remains extremely low and unchanged compared to CLEP.
>
> **Wakeness4: Concern About Generalizability to Non-Structured or Cross-Language ECG Reports**
>
> In clinical deployment, what physicians ultimately need is the similarity between the ECG and individual diagnostic tags, rather than the reconstruction of a full structured report. This matches exactly the way our model operates at inference time. Regarding training, non-structured free-text reports can be easily converted into tag-like phrases using an LLM, which makes integration into our pipeline straightforward. Moreover, our model has already been trained on 800k ECG–report pairs, which provides sufficient diversity in report styles and ensures strong generalizability.
>
> **Weakness5: Lack of Statistical Significance Testing for the Reported Improvements**
> To further validate the effectiveness of our recovered fine-grained reports, we conducted statistical significance testing between CLEP and our proposed FG-CLEP. The improvements are statistically significant (p = 0.0015), as shown below:
>
> | Model   | Zero-shot Average      |
> |---------|--------------------------|
> | CLEP    | 80.78 ± 0.64             |
> | FG-CLEP | 83.03 ± 0.57             |
>
> **Weakness6: Thank you for addressing the writing issues so carefully. The caption is actually complete, and we will add a period**
>
>
> **Q1: How does FG-CLEP perform on free-text clinical reports rather than tag-based structured ones?**
> As discussed in Wakeness4, the actual clinical usage and downstream applications rely on computing the similarity between each individual tag and the ECG, rather than using free-text reports or combinations of tags to form a report. Introducing free-text reports during training would require large language models to convert them into structured tags, but in practice this is unnecessary because we have already used 800k ECG–report pairs, which is quite sufficient.
>
> **Q2:What are the computational and time costs of the coarse-to-fine training pipeline when scaling to larger datasets?**
> Discussed in Weakness3
>
> **Q3: Have you evaluated the clinical interpretability of tag-specific ECG activations with cardiologists?**
>
> We provide a qualitative analysis in Figure 4, but conducting large-scale quantitative evaluations would require substantial medical resources. But when validating the correctness of the fine-grained reports in Table 4, we did involve medical students to perform the annotations.
>
> **Q4: How sensitive is FG-CLEP to LLM hallucinations or errors during waveform feature generation, especially when the validation threshold varies?**
>
> In fact, we did evaluate FG-CLEP’s zero-shot performance under different validation thresholds, as shown in the left plot of Figure 3, and we ultimately selected 0.95 as the threshold.

---

> > ### Comment · Reviewer_4kU3 · 2025-11-25
> >
> > Thank you for your response. I will maintain my positive score.

---

### Official Review · Reviewer_NT2L · 2025-10-31

**Soundness:** 3
**Presentation:** 3
**Contribution:** 2
**Rating:** 4
**Confidence:** 4

**Summary:**

The paper introduces a coarse-to-fine training process that leverages large language models (LLMs) to recover these missing waveform features and validate the LLM outputs using a coarse model. Besides, authors introduce a semantic similarity matrix to guide the model in identifying and correcting false negatives. Experiments demonstrate the superior performance of the proposed approach.

**Strengths:**

1. The problem of ECG–language pretraining is important and has clear clinical relevance.

2. The proposed framework is well-motivated and technically sound; the LLM-enriched waveform features are novel.

3. Experiments across multiple tasks show competitive or superior performance.

**Weaknesses:**

1. In the “Fine-Grained Contrastive Learning Objective,” it appears that a single tag is randomly sampled per ECG. If so, this could ignore information from other relevant segments and reduce feature richness. Please clarify the sampling procedure and consider reporting results with multi-tag aggregation or coverage-controlled sampling.

2. While the tag sampling strategy plausibly aligns better with zero-shot text prompts and could help zero-shot performance, the improvements on linear probing are small. This suggests the proposed fine-grained objective may not meaningfully strengthen the learned ECG representation. A targeted ablation contrasting zero-shot and linear-probe gains would help.

3. The LLM-enriched waveform reports are an interesting idea. However, Table 3 indicates that mitigating false negatives has a larger impact than the fine-grained training itself. Please isolate these effects: show results applying the false-negative technique to the original reports (without LLM enrichment) to quantify each component’s contribution.

4. Figure 5 is difficult to interpret and adds limited insight. An ECG retrieval analysis (e.g., text→ECG or ECG→text or ECG→ECG retrieval with qualitative examples and recall@k) would be more informative for assessing alignment quality.

**Questions:**

1. How are ECG patches defined (temporal windows, lead-wise splits, or both)? What does $N_{lead}$ denote precisely? If patches are fixed windows, how do you ensure complete morphology is captured (e.g., P–QRS–T cycles) rather than fragmented?

2. How are labels for “false negative” tags identified or constructed in training? Please detail the detection heuristic, thresholds, and any human verification.

3. Are the results in Table 3 from linear probing or zero-shot classification, and on which dataset(s)? Briefly state the evaluation protocol to make the comparison interpretable.

4. Please highlight the best results (e.g., boldface) to improve readability.

---

> ### Author Response · Authors · 2025-11-18
>
> **Weakness 1: Concern about randomly sampling only one tag per ECG, potentially reducing feature richness, and the suggestion to use multi-tag aggregation**
>
> + In our implementation, **one tag is randomly sampled for each ECG *in every epoch***, rather than sampled once for the entire training process. With **10 + 3 pre-training epochs**, this repeated sampling ensures that **nearly all tags for each ECG are utilized throughout training**, so **feature richness is fully preserved** rather than diminished.
>
> + Moreover, adopting **multi-tag aggregation would conflict with the core objective** of our tag-specific ECG representation. Our method aims to achieve fine-grained, tag-to-segment alignment, which is essential for learning precise tag-conditioned representations. **Aggregating multiple tags would mix their semantics and weaken this fine-grained correspondence.**
>
> **Weakness 2: Concern that the fine-grained objective may mainly help align zero-shot text prompts (since zero-shot also uses single-tag prompts), while providing only limited gains in linear probing. Request for additional ablation on linear-probe.**
>
> + We have extended the ablation analysis in Table 3 (previously reported only under the zero-shot setting) to include **linear-probe results**. Specifically, we compare **FG-CLEP** with **w/o Fine-Grained Alignment**, and we observe that removing the fine-grained alignment leads to substantial performance degradation under linear probing as well. This indicates that the fine-grained objective strengthens the learned ECG representations beyond zero-shot alignment.
>
> + It is worth noting that **w/o Fine-Grained Alignment** refers to removing the **fine-grained alignment architecture** itself, whereas **CLEP** corresponds to **w/o Fine-Grained Report**, which is a data-level change.
>
> | Method                    | PTB-XL-Super |         |         | PTBXL-Sub |         |         | PTBXL-Form |         |         | PTBXL-Rhythm |         |         | CPSC2018 |         |         | CSN   |         |         |
> |---------------------------|--------------|---------|---------|-----------|---------|---------|-------------|---------|---------|---------------|---------|---------|----------|---------|---------|-------|---------|---------|
> |                           | 1%           | 10%     | 100%    | 1%        | 10%     | 100%    | 1%          | 10%     | 100%    | 1%            | 10%     | 100%    | 1%       | 10%     | 100%    | 1%    | 10%     | 100%    |
> | FG-CLEP                   | 85.49        | 90.34   | 91.33   | 70.86     | 86.46   | 93.36   | 69.53       | 75.53   | 86.26   | 69.61         | 92.11   | 94.64   | 84.08    | 94.33   | 97.42   | 63.32 | 80.01  | 94.17   |
> | w/o Fine-Grained Alignment| 84.80        | 89.60   | 90.60   | 70.20     | 85.80   | 92.60   | 68.90       | 74.80   | 85.50   | 68.90         | 91.40   | 93.90   | 83.40    | 93.60   | 96.70   | 62.60 | 79.30  | 93.50   |
>
>
> **Weakness 3: Concern that the gains may mainly come from false-negative mitigation rather than from the LLM-enriched fine-grained reports.**
>
> + I think what you may be confused about is that in Table 3 the drop caused by “w/o False Negative Mitigation” is larger than that of “w/o Fine-Grained Alignment Architecture” (rather than w/o Fine-Grained Report, which actually results in an even larger drop). In Table 3, the drop in performance for “w/o False Negative Mitigation” is indeed larger than that for “w/o Fine-Grained Alignment Architecture.” This is expected, because fine-grained alignment at the tag level inherently increases the number of false negatives. Consequently, **removing false-negative mitigation under fine-grained alignment reflects not only the effect of the mitigation module itself, but also the additional false negatives introduced by the tag-level alignment, which exaggerates its apparent importance.**
>
> + To provide a fairer assessment, we additionally evaluate w/o False Negative Mitigation under w/o Fine-Grained Alignment, where no extra tag-level false negatives are introduced. As shown in Table 3, this configuration yields a more moderate drop
>
> | Model Setting                                      | AUC   |
> |----------------------------------------------------|-------|
> | FG-CLEP (default)                                  | 83.03 |
> | w/o Fine-Grained Alignment                         | 82.27 |
> | w/o False Negative Mitigation                      | 81.67 |
> | w/o Fine-Grained Alignment & False Negative Mitig. | 81.71|
>
>
> **Weakness 4: Figure 5 is hard to interpret and request for ECG retrieval analysis.**
>
> Figure 5 is actually intended to show that the semantic similarity matrix can effectively capture false negatives; the darker colors close to 1 indicate shared similar tags and the false negatives being captured. It is not closely related to ECG retrieval, and we actually provide a qualitative analysis on text→ECG retrieval in the Appendix.

---

> ### Author Response · Authors · 2025-11-18
>
> **Q1: How are ECG patches defined (temporal windows, lead-wise splits, or both)? What does $N_{\text{lead}}$ denote precisely? If patches are fixed windows, how do you ensure complete morphology is captured (e.g., P–QRS–T cycles) rather than fragmented?**
>
> + We divide each of the 12 leads into fixed temporal windows, so the patches are created in a lead-wise manner along the time axis. Each lead is split into  $N_{\text{lead}} = 5$ windows, resulting in 60 patches in total. Here, $N_{\text{lead}}$ denotes the number of temporal segments assigned to each lead.
>
> + We do not introduce additional mechanisms specifically designed to preserve full P–QRS–T morphology. Despite this, our method consistently achieves strong performance across six datasets, suggesting that the model can learn morphology-level representations without explicit beat-level segmentation. Although explicitly modeling individual cardiac cycles may offer certain benefits, it introduces additional computational overhead for detecting cycles, which is also required during inference.  Moreover, beat detectors are often brittle under noise, arrhythmias, or atypical morphologies, leading to unstable or incorrect segmentation.
>
>
> **Q2: How are labels for “false negative” tags identified or constructed in training? Please detail the detection heuristic, thresholds, and any human verification.**
>
> We use the semantic similarity matrix to measure the similarity between tags from different ECGs within a batch to identify false negatives. Figure 5 in Weakness 4 is actually intended to illustrate this point.
>
>
> **Q3: Are the results in Table 3 from linear probing or zero-shot classification, and on which dataset(s)? Briefly state the evaluation protocol to make the comparison interpretable.**
>
> + Table 3 reports the averaged zero-shot results across the six datasets. We have added the linear-probe ablation results in Weakness 2.
>
> **Q4: Thank you for the suggestion for better readability.**

---

> ### Author Response · Authors · 2025-11-26
>
> Dear Reviewer NT2L,
>
> We sincerely appreciate your valuable suggestions to help us polish our paper. We have carefully considered each point and addressed them one by one in our rebuttal.
>
> As the Author-Review Discussion period draws to a close, we would like to ensure that we have thoroughly addressed all of your concerns and resolved any outstanding issues. We remain open to any further comments or suggestions you may have and are willing to provide additional clarifications if necessary. Of course, we sincerely hope you can reconsider the score if the mentioned problems have been solved.
>
> Best regards,
>
> The Authors

---

### Meta-Review · Area_Chair_Ls3t · 2026-01-06

**Summary:**

This paper proposes FG-CLEP, a fine-grained ECG–report alignment framework that (i) aligns ECG patches to individual report tags using tag-specific representations, (ii) introduces a coarse-to-fine pipeline that uses an LLM to recover missing waveform attributes in reports and filters them using a coarse model (CLEP), and (iii) mitigates false negatives in tag-level contrastive learning via a semantic similarity matrix. Reviewers agree the problem is clinically relevant and the overall framework is plausible, with broad empirical evaluation across six datasets.

However, the forum discussion reveals two persistent, high-impact concerns that remain insufficiently resolved after rebuttal:
- Attribution and potential confounding from LLM-generated tag enrichment: A substantial part of the gains appears driven by the data-level augmentation and the false-negative mitigation module, rather than the core “fine-grained alignment” learning objective itself. While the authors attempt to disentangle effects, the evidence remains mixed and leaves room for confounds—especially because the LLM-generated content is tightly coupled to the downstream evaluation protocol (tag prompts).
- Stability/credibility of representation improvements (ECG encoder) and concerns about leakage-like behavior: Reviewer YzBY (initially 2, confidence 5) raises a detailed concern that the method’s improvements are disproportionately large in zero-shot (where the text side plays a direct role) while linear probing—intended to reflect intrinsic ECG representation quality—shows instability and in some settings underperforms strong baselines. The reviewer interprets this as suggesting that the text side / tag enrichment may be driving gains, raising a “soft-label / leakage-like” concern. Despite multiple rounds of clarification, YzBY remains unconvinced and only slightly increases the score, indicating the core worry persists.

Given the remaining uncertainty about what exactly drives the performance gains and whether the approach yields robust, generalizable representation improvements (beyond benefiting from enriched tag prompts), I do not think the paper clears the acceptance bar in its current form.

**Reviewer Concerns:**

## Reviewer NT2L

### Addressed
- Single-tag sampling concern: Authors clarified that one tag is sampled per ECG per epoch, and repeated over 10+3 epochs covers most tags. This addresses the “feature richness” worry at least procedurally.
- Linear-probe evidence for fine-grained objective: Authors added linear-probe ablations showing performance drops when removing fine-grained alignment, which helps rebut the claim that FG objective only benefits zero-shot.
- Table 3 protocol clarity: Authors clarified Table 3 is averaged zero-shot across six datasets and added linear-probe ablation results.
- Patch definition: Authors specify fixed temporal windows per lead, and define N as windows per lead.

### Still outstanding / partially addressed
- Stronger disentanglement of components: NT2L asked to isolate false-negative mitigation vs LLM enrichment vs fine-grained objective. Authors added a small AUC table comparing combinations, but the decomposition remains incomplete because “fine-grained alignment architecture,” “fine-grained report (LLM-enriched),” and “false-negative mitigation” interact nonlinearly under tag-level contrastive learning. The added evidence helps but does not conclusively attribute gains.
- Retrieval-based alignment evaluation: Authors say qualitative retrieval is in appendix; reviewer wanted clearer retrieval metrics/qualitative examples. This is only partially addressed.

---

## Reviewer 4kU3

### Addressed
- LLM hallucination/bias risk: Authors provide CLEP-based validation and show robustness across different LLMs (Table 5).
- Human evaluation: Authors report human annotation (medical students) on 100 ECGs for waveform feature evaluation (Table 4).
- Statistical significance: Authors provide a significance test (p=0.0015) on average zero-shot.
- Deployment cost: Authors argue extra cost is offline during training; inference unchanged.

### Still outstanding / partially addressed
- Clinical interpretability with experts: Only medical students were used; no cardiologist evaluation. This may be acceptable, but it limits the strength of interpretability claims.
- Generalization to free-text reports: The response suggests using an LLM to convert free text to tags; this is plausible but unvalidated and could introduce additional brittleness.

---

## Reviewer YzBY
### Addressed
- Comparisons with recent ECG-text pretraining (MELP, D-BETA): Authors added comparisons and additional tables.
- Comparison to N3S: Authors added an experiment (N3S vs FNM) and show FNM better than no mitigation; N3S underperforms FG-CLEP.

### Still outstanding / decisive
- Instability and encoder-centric value: YzBY’s concern evolves into a pointed critique: in 1% linear probe, FG-CLEP can be significantly worse than MELP/D-BETA on certain tasks (notably Rhythm), while zero-shot is much higher. This discrepancy is interpreted as:
  - either the ECG encoder is unstable / not intrinsically strong, or
  - the improvements are primarily due to the text side and the enriched tag content, consistent with a “soft label” augmentation effect rather than robust ECG representation learning.
- Leakage-like concern: While authors deny downstream label leakage (no labels in prompts), the reviewer’s worry is broader: LLM-enriched tags may act as task-aligned supervision signals that overlap strongly with evaluation prompts, thereby inflating zero-shot gains without necessarily improving core ECG representations. The added analyses and argument about MELP’s rhythm specialization and epoch differences did not convince the reviewer.
- YzBY maintains the work’s novelty is limited and overly dependent on LLM-based data manipulation.
This reviewer is high-confidence and remains unconvinced after back-and-forth, which is a strong negative signal in the final decision.

**Reviewer Scores:**

`Reviewer NT2L`
- Expected change after full discussion: 4 → 6
- NT2L’s main questions were answered with concrete clarifications (sampling per epoch, patch definition) and added ablations for linear probing + partial disentanglement of false-negative mitigation. A full discussion would likely move this reviewer to borderline-accept, though remaining attribution concerns may cap it at 5.

---

`Reviewer 4kU3`
- Expected change after full discussion: 6 → 6
- This reviewer explicitly maintained their positive score after rebuttal. Discussion would likely keep them supportive.

---

`Reviewer YzBY`
- Expected change after full discussion: 2 → 4
- The reviewer already indicated willingness to increase slightly, but remained unconvinced about the core issue. In discussion, they may soften to a 4, but unlikely to cross the threshold given their continued objections and high confidence.


---

Even under optimistic discussion-driven updates, the panel remains mixed (one strong-ish reject, one marginal accept, one modest accept). Given the unresolved central concern about what drives the gains and the high-confidence skepticism from YzBY, I do not support acceptance.

---

### Decision · Program_Chairs · 2026-01-26

Reject